# Multi-User Computation Offloading and Resource Allocation Algorithm in a Vehicular Edge Network

**DOI:** 10.3390/s24072205

**Published:** 2024-03-29

**Authors:** Xiangyan Liu, Jianhong Zheng, Meng Zhang, Yang Li, Rui Wang, Yun He

**Affiliations:** 1School of Communications and Information Engineering, Chongqing University of Posts and Telecommunications, Chongqing 400065, China; zhengjh@cqupt.edu.cn (J.Z.); d190101012@stu.cqupt.edu.cn (R.W.); heyun@cqupt.edu.cn (Y.H.); 2State Key Laboratory of Block Chain and Data Security, Zhejiang University, Hangzhou 310058, China; zhangmengyang@zju.edu.cn; 3Cyberspace Security Key Laboratory of Sichuan Province, Chengdu 610043, China; liyang.yalee@gmail.com; 4Department of Electronic Communication Engineering, Yuxi Normal University, Yuxi 653100, China

**Keywords:** Vehicular Edge Computing Network (VECN), computation offloading, resource allocation, deep reinforcement learning

## Abstract

In Vehicular Edge Computing Network (VECN) scenarios, the mobility of vehicles causes the uncertainty of channel state information, which makes it difficult to guarantee the Quality of Service (QoS) in the process of computation offloading and the resource allocation of a Vehicular Edge Computing Server (VECS). A multi-user computation offloading and resource allocation optimization model and a computation offloading and resource allocation algorithm based on the Deep Deterministic Policy Gradient (DDPG) are proposed to address this problem. Firstly, the problem is modeled as a Mixed Integer Nonlinear Programming (MINLP) problem according to the optimization objective of minimizing the total system delay. Then, in response to the large state space and the coexistence of discrete and continuous variables in the action space, a reinforcement learning algorithm based on DDPG is proposed. Finally, the proposed method is used to solve the problem and compared with the other three benchmark schemes. Compared with the baseline algorithms, the proposed scheme can effectively select the task offloading mode and reasonably allocate VECS computing resources, ensure the QoS of task execution, and have a certain stability and scalability. Simulation results show that the total completion time of the proposed scheme can be reduced by 24–29% compared with the existing state-of-the-art techniques.

## 1. Introduction

The emergence of various intelligent on-vehicle applications in the Internet of Vehicles (IoV), such as autonomous driving, online games, augmented reality, intelligent guidance of traffic behavior, and voice-based dynamic human-vehicle interaction, makes resource-constrained vehicles face significant challenges in supporting these intelligent services [1,2,3,4,5]. Vehicular Edge Computing Networks (VECNs) extend computation capability to the edge of the wireless network by providing additional computation resources close to mobile vehicles, which can ease the burden on vehicles. Moreover, VECNs make it possible to take full advantage of ubiquitous computation resources in the system. Computing offloading is used to realize computation-intensive and delay-sensitive applications processed in the ubiquitous computation resources, which frees Task Vehicles (TaVs) from complex tasks, helps to reduce service delay, effectively alleviates the problem of limited computation capability of TaVs, and provides better Quality of Service (QoS) for vehicle users [6,7]. However, the mobility of vehicles and the diversity of edge computing nodes and vehicle offloading modes bring challenges to task offloading services [8].

Computation tasks can be offloaded to Service Vehicles (SeVs) via vehicle-to-vehicle (V2V) links to use computation resources in the system. It can also be offloaded to a Vehicular Edge Computing Server (VECS) connected to a Road Side Unit (RSU) or a Base Station (BS) via a vehicle-to-infrastructure (V2I) link. Thus, a vehicle offloading mode mainly includes Local execution mode (Loc), Local + SeV execution mode (Loc + SeV), Local + VECS execution mode (Loc + Edge), and Local + SeV + VECS execution mode (Loc + Sev + Edge).

The remainder of this paper is organized as follows: In Section 2, related works are discussed. The system model and problem formulation are formulated in Section 3. The multi-user computing offloading and resource allocation method based on Deep Deterministic Policy Gradient (DDPG) is given in Section 4. We conduct the simulation results and analysis of the proposed algorithm in Section 5. The conclusion and future work are given in Section 6.

## 2. Related Work

V2V links are used to offload tasks to SeVs [9,10]. Platooning vehicles are considered in work [9], where platoon members can only communicate with the platoon leader and use the resources via V2V communication links. More factors are considered in [10] when SeVs are selected, such as the caching factor, energy factor, and location factor of vehicles to offload non-real-time traffic to V2V networks.

V2I links are used to offload tasks to VECS in the way of partial offloading [11,12,13,14,15,16]; namely, parts of tasks are processed locally, and others are offloaded to VECS. The environmental settings and optimization objectives distinguish these works. TaVs should pay for the services provided by VECS in [14]. Analytical offloading schemes for some special VECNs are proposed in [15], including the cases of one TaV with one VECS, one TaV with two VECSs, and two TaVs with one VECS. The mobility of vehicles is not considered in some works [11,12,13,14,15]. However, the mobility of TaVs while TaVs move in a random direction at a specific rate is considered in [16].

Both V2V and V2I links are leveraged to offload tasks to SeVs and VECSs, which can fully use the system’s ubiquitous computation resources [17,18,19,20]. VECSs deployed at RSUs are regarded as fixed VECSs, while the mobile vehicles are regarded as mobile VECSs. The two types of VECSs cooperate to provide additional computing resources for TaVs [17]. Except for mobile vehicles, parked vehicles also can be treated as SeVs [18,19]. Based on this, the work in [18] proposes a dynamic pricing strategy to maximize the revenue of the computing service provider. In contrast, the work in [19] organizes RSUs and roadside parked vehicles into parking clusters to make up for the computation resource bottleneck caused by insufficient infrastructure construction. Furthermore, considering the importance of the matching between TaVs and corresponding processing terminals, the authors in [20] propose a four-lane dual carriageway model to simulate the urban traffic environment and use the Kuhn-Munkras algorithm to realize the matching of TaVs and service providers.

Some of the literature adopts traditional methods for computing offloading [21,22,23,24,25]. For instance, a queue-based improved multi-objective particle swarm optimization algorithm to solve the problem of multi-dependent task offloading in multi-access edge computing is proposed in [21]. The author in [22] divides and conquers the goal into two phases: VECS selection and offloading decision. For the VECSs selection phases, TaVs are grouped into one BS, considering their physical distance and workload. After VECS selection, the original problem is divided into parallel multi-user-to-one-server offloading decision subproblems and a distributed offloading strategy based on a binary-coded genetic algorithm is used to obtain an adaptive offloading decision. Considering the heterogeneity of communication modes and computing capabilities of network computing points in ubiquitous networks, a distributed multi-hop computing task offloading framework based on an improved genetic algorithm is proposed in [23] so that tasks could be recursively offloaded among computing points in the ubiquitous network. Benders decomposition technology is used to realize task offloading in [24]. The author in [25] considers the quasi-static channel model during task offloading, wherein the channel remains constant during the offloading period but may change during different offloading periods; a two-stage Stalberg game is then used to solve the optimization objective.

Computing offloading via V2V communication and V2I communication can make full use of the ubiquitous computing resources of the system and improve the performance of mobile edge computing [26]. However, due to the mobility of vehicles and dynamic wireless channel conditions, the formulation of computing offloading strategies has high-dimensional and time-varying characteristics. Most of the optimization-based computation offloading schemes lack the ability to adapt to dynamic environments. Fortunately, deep reinforcement learning in artificial intelligence can solve such high-dimensional time-varying feature problems with limited and inaccurate information [27,28]. Deep reinforcement learning algorithms for task offloading management are used in some of the literature [29,30,31,32]. Based on this, the computation tasks of TaVs are offloaded to edge vehicles and cloud networks to acquire more computation resources [29]. The problem of computation offloading and resource allocation for tasks offloading to VECS through V2I links is addressed in [30]. Tasks are offloaded hierarchically in [31]. A vehicle may have multiple tasks, and the author in [32] considers offloading these tasks to multiple vehicles, nearby pedestrians that use mobile phones or tablets, other vehicles that can provide computing services, and VECSs. The characteristics, pros, and cons revealed in the recent research are provided in Table 1. For simplicity, $M^{0},M^{1},M^{2},M^{3}$ are used to denote the four modes of task execution, where the $M^{0}$, $M^{1}$, $M^{2}$, and $M^{3}$ modes represent Loc mode, Loc + Sev mode, Loc + Edge mode, and Loc + Sev + Edge mode, respectively.

As seen in Table 1, this literature is based on the three execution modes, $M^{1}$, $M^{2}$, and $M^{3}$, which all involve the local execution mode $M^{0}$. Most of the research in the $M^{1}$ and $M^{2}$ modes adopts the partial offloading mode [9,10,14,15,16,25,29], and most of the research in the $M^{3}$ mode adopts the 0–1 offloading mode [19,20,23,31,32]. Furthermore, we have studied the use of V2V and V2I links to extend the system’s computing resources [33,34]. However, the computing offloading and the resource allocation of VECSs in dynamic environments have not been fully considered. Based on this, this paper comprehensively considers TaV’s preference for the Loc mode, Loc + Sev mode, Loc + Edge mode, and Loc + Sev + Edge mode in a dynamic environment and the impact of task offloading and resource allocation on offloading delay. The computation offloading and resource allocation problem is modeled as a Mixed Integer Nonlinear Programming (MINLP) problem. Then, considering the advantage of DDPG for environmental dynamics, a method based on DDPG is proposed; namely, the multi-user computation offloading and resource allocation scheme (MCORA). The main contributions of this paper are summarized as follows:To solve the problem of task execution time being difficult to acquire because of the mobility of vehicles and the dynamic of channel state information, a computing offloading and resource allocation optimization scheme is proposed for multiple TaV, which adopts the best mode from four execution modes; namely, Loc mode, Loc + Sev mode, Loc + Edge mode, and Loc + Sev + Edge mode. Leveraging these four modes, we can analyze the complex task execution process more simply and acquire the task execution time.To minimize the total task execution time by choosing the adaptive mode and allocating the computation resources of a VECS, the optimization objective is established according to the delay of task execution, and then the computing offloading mode chosen with the resource allocation problem is transformed into a MINLP problem. This can be described as a Markon Decision Processes (MDP), and the MCORA algorithm is proposed to solve it.To solve the non-convexity and the discontinuity of the offloading mode selection and the resource allocation of this problem, DDPG is considered because it can deal with continuous and discontinuous actions, so the MCORA scheme is based on DDPG.To verify the effectiveness of our scheme, three baseline schemes are compared with our scheme; namely, Offloading in Loc + Sev mode (OLSM) [10], Offloading in Loc + Edge mode (OLEM) [14], and Offloading in Random Mode (ORM). Simulation results show that compared with the existing schemes, the proposed scheme can significantly reduce the delay of computation offloading and resource allocation.

## 3. System Model and Problem Formulation

### 3.1. System Model

Four communication modes can be adopted; namely, Loc mode, Loc + SeV mode, Loc + Edge mode, and Loc + SeV + Edge mode. The architecture of the VECNs is shown in Figure 1.

Four modes are depicted in Figure 1. It was Loc mode when TaV executed all tasks locally, Loc + SeV mode when TaV executed some tasks locally and offloaded the others to SeV, Loc + Edge mode when TaV executed some tasks locally and offloaded the others to VECS, and Loc + SeV + Edge mode when tasks were executed in the three terminals.

Set $N=\{V_{1},V_{2},\cdots ,V_{n}\cdots ,V_{N}\}$ represents *N* TaVs randomly distributed on an urban traffic environment, and all TaVs are connected to a BS located in the center. A VECS is deployed at the BS. TaVn has a task $I_{n}=\left\{D_{n},App_{n}\right\}$ to be processed, where $D_{n}$ (in bits) represents task sizes, and $App_{n}$ (in CPU cycles/bit) represents the processing density of TaVn. Based on this, $C_{n}=D_{n}App_{n}$ represents the CPU resources required to complete this task. These tasks are all arbitrarily divisible, and their maximum processing delay is $t_{\max}$, which is also the processing period of these tasks. We divide the time into *T* equal time slots T={1,2,⋯,t,⋯,T}, and the time in each time slot is $\tau =t_{\max}/T$. At any time slot, the task can be executed in any execution mode.

### 3.2. Communication Model

Based on the available literature [16], this paper further considers the relative position and the task offloading between TaVs and SeVs. Then, the communication model is established as follows: TaVn and SeVn are moving at a certain speed $v_{n}^{\mathrm{tav}}$ and $v_{n}^{\mathrm{sev}}$, respectively, assuming that the distance between them changes in a uniformly distributed range $d_{\mathrm{tav}}^{\mathrm{sev}}$. Moreover, V2I links and V2V links are all adopting orthogonal frequency division multiplexing technology [20]. Channel power gains of the V2V link from TaVn to SeVn and the V2I link from TaVn to VECS are denoted as $g_{n,t}^{v2v/v2i}=\alpha_{n,t}^{v2v/v2i}\cdot h_{n,t}^{v2v/v2i}$, where $\alpha_{n,t}^{v2v/v2i}$ and $h_{n,t}^{v2v/v2i}$ represent the large-scale fading and small-scale fading of the V2V links and V2I links, respectively. The small-scale fading is exponentially distributed. $\alpha_{n,t}^{v2v/v2i}$ includes path loss and shadow fading. The path losses of the V2V links and V2I links are calculated as follows [35]:(1)$$\begin{array}{c} PL_{v2v}^{t}\left(d_{\mathrm{tav}}^{\mathrm{sev}}\right)=\left\{\begin{array}{c} 22.7\log_{10}3+41+20\log_{10}\left(\frac{freq}{5}\right),d_{\mathrm{tav}}^{\mathrm{sev}}\leq 3\\ 22.7\log_{10}\left(d_{\mathrm{tav}}^{\mathrm{sev}}\right)+41+20\log_{10}\left(\frac{freq}{5}\right),d_{\mathrm{tav}}^{\mathrm{sev}}\leq \frac{4freq{(H^{\mathrm{veh}}-1)}^{2}}{c}\\ 40\log_{10}\left(d_{\mathrm{tav}}^{\mathrm{sev}}\right)+9.45-17.3\log_{10}({\left(H^{\mathrm{veh}}-1\right)}^{2})+2.7\log10(\frac{freq}{5}),\mathrm{else} \end{array}\right. \end{array},$$
and
(2)$${\mathrm{PL}}_{v2i}^{t}\left(d_{n}^{\mathrm{edg}}\right)=128.1+37.6\log_{10}(\frac{d_{n}^{\mathrm{edg}}}{1000}),$$
where $H^{\mathrm{veh}}$ represents the antenna height of the vehicle, freq denotes the carrier frequency, and $d_{n}^{\mathrm{edg}}$ denotes the distance between TaVn and VECS. The updated formula of shadow fading is as follows [35]:(3)$$S_{t}^{v2v/v2i}=S_{t-1}^{v2v/v2i}\cdot e^{-\frac{\left(\Delta_{\mathrm{tav}}^{t}+\Delta_{\mathrm{sev}}^{t}\right)}{10}}+S^{v2v/v2i}\cdot \sqrt{1-e^{-\frac{2(\Delta_{\mathrm{tav}}^{t}+\Delta_{\mathrm{sev}}^{t})}{10}}},$$
where $\Delta_{\mathrm{tav}}^{t}$ and $\Delta_{\mathrm{sev}}^{t}$ denote the distance traveled by TaV and SeV in the *t*th time slot. Then, the data transmission rate of the V2V links and V2I links can be expressed as follows:(4)$$R_{n,t}^{v2v/v2i}\;=\;B\log_{2}\left(1+\gamma_{n,t}^{v2v/v2i}\right),$$
where $\gamma_{n,t}^{v2v/v2i}=P_{n}g_{n,t}^{v2v/v2i}/\delta^{2}$. $P_{n}$ represents the transmission power of TaVn.

### 3.3. Mode Selection and Task Offloading Delay Computing in Edge Networks

At time slot *t*, if TaVn chooses to execute tasks locally, the number of bits that can be processed can be expressed as follows:(5)$$U_{n,t}^{\mathrm{loc}}=\frac{\tau \cdot f_{n}^{\mathrm{loc}}}{App_{n}},$$
where $f_{n}^{\mathrm{loc}}$ denotes the processing capacity of TaVn. If TaVn chooses to execute tasks at SeVn, the task needs to be offloaded to SeVn at first, and the task in time slot *t* includes not only the time to transmit to SeVn but also the time to execute the task at SeVn. Let $U_{n,t}^{\mathrm{sev}}$ represent the number of bits that can be completed in time slot *t*. Then, according to the transmission time $sev_{\mathrm{tr}}=U_{n,t}^{\mathrm{sev}}/R_{n}^{v2v}$, the computation time $sev_{e}\;=\;U_{n,t}^{\mathrm{sev}}\cdot App_{n}/f_{n}^{\mathrm{sev}}$, and the equality $sev_{\mathrm{tr}}+sev_{e}\;=\;\tau$, $U_{n,t}^{\mathrm{sev}}$ can be obtained in time slot *t* as follows:(6)$$U_{n,t}^{\mathrm{sev}}=\frac{\tau \cdot R_{n}^{v2v}\cdot f_{n}^{\mathrm{sev}}}{f_{n}^{\mathrm{sev}}+App_{n}\cdot R_{n}^{v2v}}.$$

Similarly, when tasks of a TaV are selected to be executed at VECS through the V2I link, according to the transmission time $edg_{\mathrm{tr}}=U_{n,t}^{\mathrm{edg}}/R_{n}^{v2I}$, the computing time $edg_{e}\;=\;U_{n,t}^{\mathrm{edg}}\cdot App_{n}/\left(\rho_{n}^{t}\cdot F^{\mathrm{edg}}\right)$, and $edg_{\mathrm{tr}}+edg_{e}\;=\;\tau$, the number of bits that can be processed in time slot *t* is obtained as follows:(7)$$U_{n,t}^{\mathrm{edg}}=\frac{\tau \cdot R_{n}^{v2I}\cdot \rho_{n}^{t}\cdot F^{\mathrm{edg}}}{\rho_{n}^{t}\cdot F^{\mathrm{edg}}+App_{n}\cdot R_{n}^{v2I}}.$$

$M_{n}\in \left\{M^{0},M^{1},M^{2},M^{3}\right\}$ is used to denote the four modes of task execution. When TaVn chooses $M^{0}$ mode to execute tasks, the number of bits that can be processed in time slot *t* is equal to the number of bits that can be executed locally:(8)$$U_{\mathrm{total},n}^{t}=U_{n,t}^{\mathrm{loc}}=\frac{\tau \cdot f_{n}^{\mathrm{loc}}}{App_{n}}.$$

When TaVn chooses mode $M^{1}$ to execute tasks, the number of bits that can be completed in time slot *t* is equal to the sum of the number of bits that can be executed locally and in SeVn:(9)$$U_{\mathrm{total},n}^{t}=U_{n,t}^{\mathrm{loc}}+U_{n,t}^{\mathrm{sev}}=\frac{\tau \cdot f_{n}^{\mathrm{loc}}}{{App}_{n}}+\frac{\tau \cdot R_{n}^{v2v}\cdot f_{n}^{\mathrm{sev}}}{f_{n}^{\mathrm{sev}}+{App}_{n}\cdot R_{n}^{v2v}}.$$

Similarly, in $M^{2}$ mode, the number of bits that can be processed is the sum of the tasks that can be processed locally and by VECS:(10)$$U_{n,t}^{\mathrm{total}}=U_{n,t}^{\mathrm{loc}}+U_{n,t}^{\mathrm{edg}}=\frac{\tau \cdot f_{n}^{\mathrm{loc}}}{App_{n}}+\frac{\tau \cdot R_{n}^{v2I}\cdot \rho_{n}^{t}\cdot F^{\mathrm{edg}}}{\rho_{n}^{t}\cdot F^{edg}+App_{n}\cdot R_{n}^{v2I}}.$$

In $M^{3}$ mode, the number of bits that can be processed is the sum of the bits of the three terminals:(11)$$U_{n,t}^{\mathrm{total}}=U_{n,t}^{\mathrm{loc}}+U_{n,t}^{\mathrm{sev}}+U_{n,t}^{\mathrm{edg}}=\frac{\tau \cdot f_{n}^{\mathrm{loc}}}{{App}_{n}}+\frac{\tau \cdot R_{n}^{v2v}\cdot f_{n}^{\mathrm{sev}}}{f_{n}^{\mathrm{sev}}+{App}_{n}\cdot R_{n}^{v2v}}+\frac{\tau \cdot R_{n}^{v2I}\cdot \rho_{n}^{t}\cdot F^{\mathrm{edg}}}{\rho_{n}^{t}\cdot F^{edg}+{App}_{n}\cdot R_{n}^{v2I}}.$$

Since a TaV must choose an execution mode to execute its tasks as long as it is not completed, the completion time $t_{n}^{\mathrm{total}}$ of TaVn’s tasks can be expressed as the minimum number of time slots spent for the cumulative maximum execution bits of its tasks:(12)$$t_{n}^{\mathrm{total}}=t\cdot \tau ,\;t=\min_{t}\max\sum_{t=1}^{T}U_{n,t}^{\mathrm{total}}.$$

### 3.4. Problem Formulation

The execution time of all completed tasks in time slot *t* can be expressed as follows:(13)$$P_{t}\left(M(t),\rho \left(t\right)\right)=\sum_{t=1}^{T}t_{n}^{\mathrm{total}}$$
where $M(t)=\{M_{n}(t)|n\in N\}$ represents the mode chosen by TaVn in time slot *t*. $\rho \left(t\right)=\left\{\rho_{n}(t)|n\in N\right\}$ represents the proportion of computation resources allocated to TaVn by VECS in time slot *t*. Our goal is to minimize the execution time of all tasks:$$\begin{array}{cc} P & \min_{M(t),\rho (t)}P_{t}\left(M(t),\rho \left(t\right)\right)=\sum_{t=1}^{T}\min_{M(t),\rho (t)}t_{n}^{\mathrm{total}}\\ s.t. & C1:M_{n}(t)\in \left\{M^{0},M^{1},M^{2},M^{3}\right\},n\in N\\ \; & C2:0\leq \sum_{n\in N}\rho_{n}(t)\leq 1\\ \; & C3:0\leq \rho_{n}(t)\leq 1,n\in N\\ \; & C4:t_{total,n}\leq t_{\max},n\in N \end{array}$$
where C1 denotes the mode chosen by TaVn, which is one of the four modes, and C2 indicates that the VECS resources allocated to all TaVs do not exceed the total computation resources of the VECS. C3 limits the proportion of the VECS’s computation resources allocated to each TaV. C4 means that the execution time does not exceed the maximum delay of TaVn.

## 4. Multi-User Computing Offloading and Resource Allocation Method Based on DDPG

In order to solve the formulated objective, we propose a DDPG-based computing offloading and resource allocation scheme. Considering the actual VECNs environment, the objective P is described as MDP. Then, the DDPG algorithm is designed to solve it, and the key issues are the normalization of states and actions and the design of the reward function. Finally, the task offloading algorithm based on DDPG was used to solve the optimization objective.

### 4.1. Markov Decision Processes for Mode Selection and Computing Offloading

MDP is a mathematical framework for describing sequential decision-making problems with stochastic properties [36]. A Markov model can be represented as a quadruple (S,A,P,R). The elements inside represent the set of states, the set of actions, the states’ transition probability, and the immediate reward function for performing the actions.

(1) State space. System state $s_{t}\in S$ can be expressed as follows:(14)$$s_{t}=\left(R(t),O(t),Ti\right),$$
where (a) $R(t)=\{[R_{1}^{v2v}(t)\ldots ,R_{n}^{v2v}(t),\ldots ,R_{N}^{v2v}(t)],[R_{1}^{v2i}(t)\ldots ,R_{n}^{v2i}(t),\ldots ,R_{N}^{v2i}(t)]\}$ denotes the data transmission rate of V2V/V2I links at time *t*; (b) $O(t)=\{[O_{1}(t)\ldots ,O_{n}(t),\ldots ,O_{N}(t)]\}$ represents the proportion of TaVn’s tasks remaining to be processed; (c) Ti={Ti(t)} denotes the remaining processing time.

(2) Action space. $a_{t}\in A$ can be expressed as follows:(15)$$a_{t}=(M(t),\rho (t)),$$
where (a) $M(t)=\{[M_{1}^{m}(t),\ldots ,M_{N}^{m}(t)]\}$ denotes the mode selected by each TaVs in time slot *t*; (b) $\rho \left(t\right)=\{[\rho_{1}(t),\ldots ,\rho_{N}(t)]\}$ denotes the proportion of computation resources allocated to TaVn by VECS at time slot *t*.

(3) Reward function.
(16)$$R_{t}^{\mathrm{im}}\left(s_{t},a_{t}\right)=\left\{\begin{array}{c} 1\;\mathrm{if}\;\pi (t)=N,\\ 1-(P_{t}\left(M(t),\rho \left(t\right)\right))\;\mathrm{else}, \end{array}\right.$$
where π(t) represents the number of tasks completed in time slot *t*. If all tasks are completed, the reward is assigned to 1 immediately; otherwise, the average remaining available time is assigned to the reward function.

### 4.2. DDPG-Driven Computation Resource Offloading and Resource Allocation Strategies

The DDPG-based deep reinforcement learning algorithm is used to solve the joint computing offloading and resource allocation problem. As shown in Figure 2, the algorithm includes three modules: main network, target network, and experience replay memory. The policy of the main network is to produce action $a_{t}$ based on current state $s_{t}$. The main network includes two parts, the main actor depth neural network (DNN) $\pi (s_{t}|\theta_{\pi })$ and the main critic DNN $Q(s_{t},a_{t}|\theta_{Q})$. The target network, aiming to train the network in the target, has the same structure as the main network. The parameters can be expressed as $\pi^{\prime }\left(s_{t}|{\theta^{\prime }}_{\pi }\right)$ and $Q^{\prime }\left(s_{t},a_{t}|{\theta^{\prime }}_{Q}\right)$. The experience replay memory is used to store the resulting experience tuples.

(1) Main actor DNN training. The explored policy can be defined as a function with parameter $\theta_{\pi }$, which maps the current state to an action ${\hat{a}}_{t}=\pi \left(s_{t}|\theta_{\pi }\right)$, where ${\hat{a}}_{t}$ is obtained by mapping, and $\pi (s_{t}|\theta_{\pi })$ is the model selection and computation resource allocation policy obtained by the exploration of actor DNN. The added noise $n_{t}$ follows Gaussian distribution $n_{t}∼\left(\mu_{t},\sigma_{t}^{2}\right)$. Then, the refactoring action can be expressed as follows:(17)$$a_{t}=\mathrm{clip}\left(\pi \left(s_{t}|\theta_{\pi }\right)+n_{t},a_{\mathrm{low}},a_{\mathrm{high}}\right),$$
where the clip function limits the range of action values to $a_{\mathrm{low}}$ and $a_{\mathrm{high}}$, and the main actor DNN uses sampled policy gradients to update the network parameters:(18)$$\nabla_{\theta_{\pi }}J\approx \left[\nabla_{a}Q(s_{t},a_{t}|\theta_{Q})\nabla_{\theta_{\pi }}\pi \left(s_{t}|\theta_{\pi }\right)\right],$$
where $Q(s_{t},a_{t}|\theta_{Q})$ is an action-value function. At each step of the training process, $\theta_{\pi }$ is updated by a batch of experience $<s_{t},a_{t},R_{t}^{\mathrm{im}},s_{t+1}>$:(19)$$\theta_{\pi }=\theta_{\pi }-\frac{\alpha_{\pi }}{V}\sum_{t=1}^{V}\left[\nabla_{a}Q(s_{t},a_{t}|\theta_{Q})\nabla_{\theta_{\pi }}\pi \left(s_{t}|\theta_{\pi }\right)\right],$$
where $\alpha_{\pi }$ represents the learning rate of the main actor DNN.

(2) Main critic DNN training. The main critic DNN evaluates the performance of the selected action based on the action-value function. The action-value function is computed based on the Bellman optimality equation, which can be expressed as follows:(20)$$Q(s_{t},a_{t}|\theta_{Q})=\left[R_{t}^{\mathrm{im}}\left(s_{t},a_{t}\right)+\varepsilon Q(s_{t+1},\pi \left(s_{t+1}\right)|\theta_{Q})\right],$$
where the main critic DNN considering the current state $s_{t}$ and the next state $s_{t+1}$ is used to calculate each state-action value $Q(s_{t},a_{t}|\theta_{Q})$. The main critic DNN updates the network parameters $\theta_{Q}$ by minimizing the loss function $Ls(\theta_{Q})$:(21)$$Ls\left(\theta_{Q}\right)=\left[{\left(y_{t}-Q\left(s_{t},a_{t}|\theta_{Q}\right)\right)}^{2}\right],$$
where $y_{t}$ is the target value, which can be expressed as follows:(22)$$y_{t}=R_{t}^{\mathrm{im}}\left(s_{t},a_{t}\right)+\varepsilon Q^{\prime }\left(s_{t+1},\pi^{\prime }\left(s_{t+1}|\theta_{\pi }^{\prime }\right)|\theta_{Q}^{\prime }\right),$$
$Q^{\prime }\left(s_{t+1},\pi^{\prime }\left(s_{t+1}|\theta_{\pi }^{\prime }\right)|\theta_{Q}^{\prime }\right)$ is obtained by the target network which is the network with parameters $\theta_{\pi }^{\prime }$ and $\theta_{Q}^{\prime }$. The gradient computation of $Ls(\theta_{Q})$ is expressed as follows:(23)$$\nabla_{\theta_{Q}}\mathrm{Ls}=\left[2\left(y_{t}-Q\left(s_{t},a_{t}|\theta_{Q}\right)\right)\nabla_{\theta_{Q}}Q(s_{t},a_{t})\right].$$

In each training step, $\theta_{Q}$ is updated by a batch of experience $<s_{t},a_{t},R_{t}^{\mathrm{im}},s_{t+1}>$ as follows:(24)$$\theta_{Q}=\theta_{Q}-\frac{\alpha_{Q}}{V}\sum_{t=1}^{V}\left[2\left(y_{t}-Q\left(s_{t},a_{t}|\theta_{Q}\right)\right)\nabla_{\theta_{Q}}Q(s_{t},a_{t})\right],$$
where $\alpha_{Q}$ represents the learning rate of the main actor DNN.

(3) Target network training. The target network can be regarded as an older main network version with different parameters $\theta_{\pi }^{\prime }$ and $\theta_{Q}^{\prime }$. In each iteration, the parameters $\theta_{\pi }^{\prime }$ and $\theta_{Q}^{\prime }$ are updated according to (Equation 25):(25)$$\begin{array}{c} \theta_{\pi }^{\prime }=\omega \theta_{\pi }+(1-\omega )\theta_{\pi }^{\prime }\\ \theta_{Q}^{\prime }=\omega \theta_{Q}+(1-\omega )\theta_{Q}^{\prime } \end{array},$$
where ω∈[0,1].

The computation offloading and resource allocation algorithm based on DDPG is shown in Algorithm 1. Firstly, parameter $\theta_{\pi }$ is used to initialize the computation offloading and resource allocation strategy $\pi (s|\theta_{\pi })$ of main actor DNN, and parameter $\theta_{Q}$ is used to initialize the action-value function of critic DNN $Q(s_{t},a_{t}|\theta_{Q})$. The parameters $\theta_{\pi }^{\prime }$ and $\theta_{Q}^{\prime }$ of the target network are initialized at the same time. Then, the main actor DNN generated action $a_{t}$ according to the current policy $\pi (s|\theta_{\pi })$ and state $s_{t}$. Based on the observed reward $R_{t}^{\mathrm{im}}\left(s_{t},a_{t}\right)$ and the next state $s_{t+1}$, the tuple $<s_{t},a_{t},R^{\mathrm{im}}\left(s_{t},a_{t}\right),s_{t+1}>$ is constructed and stored in an experience replay memory. The memory is stored in a first-in-first-out manner, and if the memory is about to overflow, the oldest experience will be deleted and updated to the latest experience. Based on the mini-batch technique, the algorithm updates the DNN network of the main critic DNN by minimizing the function $Ls(\theta_{Q})$ and updates the main actor DNN by using the sampled policy gradient. After a period of training, the parameters of the target network are updated according to (Equation 25).
**Algorithm 1** Multi-user computation offloading and resource allocation algorithm**Initialization:**1. Leverage parameters $\theta_{\pi }$ and $\theta_{Q}$ to initialize $\pi (s|\theta_{\pi })$ and $Q(s,a|\theta_{Q})$;2. Leverage parameters ${\theta^{\prime }}_{\pi }\leftarrow \theta_{\pi }$ and ${\theta^{\prime }}_{Q}\leftarrow \theta_{Q}$ to initialize $\pi^{\prime }\left(s|{\theta^{\prime }}_{\pi }\right)$$Q^{\prime }\left(s,a|{\theta^{\prime }}_{Q}\right)$;3. Initialize experience replay memory;**for** each episode **do:** Initialize system environment setup; **for** each time slot *t* **do:**     Acquire action $a_{t}$ according to (Equation 17);     Obtain immediate reward $R^{\mathrm{im}}\left(s_{t},a_{t}\right)$ with (Equation 16) and accumulated reward, update next state $s_{t+1}$;     **if** experience replay memory is not full **do:**      Store tuple $<s_{t},a_{t},R^{\mathrm{im}}\left(s_{t},a_{t}\right),s_{t+1}>$ into experience replay memory;     **else:**      A batch tuple *V* is randomly drawn from the experience replay memory;      The target value $y_{t}$ is calculated based on (Equation 22);      Parameters $\theta_{Q}$ are updated by minimizing the loss function based on (Equation 21);      Parameters $\theta_{\pi }$ are updated according to the sampled policy gradient based on (Equation 18);      Parameters $\theta_{\pi }^{\prime }$ and $\theta_{Q}^{\prime }$ are updated based on (Equation 25);     **End if** **End for****End for** 

## 5. Simulation Results and Analysis

### 5.1. Simulation Environment

The experiment was carried out on the Windows10 operating system with the processor Intel Core i7-6700 CPU @ 3.40 GHz (Santa Clara, CA, USA), while the software used was Python3.7.9 and TensorFlow1.15.0. The urban IoV simulator, including the vehicle, lane, and wireless communication network model defined in Appendix A of 3GPP TR 36.885 [35] is adopted. The main simulation parameters are shown in Table 2. The actor and critic networks of the DDPG agent both consist of three fully connected hidden layers consisting of 64, 16, and 4 neurons, respectively. ReLU is used as the activation function, and Adam is used as the optimizer to train and update the weights of the neural network iteratively. The algorithm was trained for a total of 2000 episodes, and the exploration probability was annealed by linear annealing algorithm from 1 at the beginning to 0.01 at 1600 episodes, and then remained unchanged in the following training steps [37]. Unless otherwise specified, the simulation parameters in this chapter are executed according to Table 2, and the results are the average values of the last 100 episodes.

### 5.2. Baseline Algorithms

OLSM [10]: TaVs choose to offload part of the tasks to corresponding SeVs via V2V links.OLEM [14]: TaVs choose to offload part of the tasks to the edge server via V2I links.ORM: TaVs choose the offloading mode randomly.

### 5.3. Simulation Results

In this section, the convergence of the proposed algorithm is first analyzed. Then, the cumulative reward and performance of the proposed algorithm are verified and compared with the baseline algorithms in four aspects: the number of TaVs, task size, required computation resources per bit, and the computing capability of vehicles and VECS.

Figure 3 shows the changing trend of different algorithms’ rewards with the number of iterations. When the number of iterations is 500, the proposed MCORA and OLSM algorithms converge. In contrast, the cumulative rewards of the OLEM and ORM algorithms are relatively stable throughout the process. Due to the change in network topology and channel fading caused by vehicle mobility, the fluctuation of TaVs’ task size and processing density, and the change in system computing capability, the convergence value of cumulative rewards will fluctuate. It is seen that MCORA has the best cumulative reward.

Figure 4 shows the cumulative rewards of different algorithms as the number of TaVs changes. As the number of TaVs increases, the cumulative rewards of all algorithms decrease. Due to the limitation of VECS resources, the cumulative reward of the OLEM algorithm decreases sharply, and the decreasing trend of the ORM algorithm is faster than that of the OLSM algorithm because part tasks in the ORM algorithm choose Loc + Edge mode. Moreover, the decreasing trend of the cumulative reward of the MCORA algorithm and the OLSM algorithm is relatively stable, and the cumulative reward of the MCORA algorithm is always the largest.

Figure 5 shows the performance of different algorithms when the number of TaVs increases, and the performance comparison mainly includes the loss rate, the total completion time of tasks, and the maximum task completion time. The loss rate is defined as the ratio of tasks that cannot be completed with limited delay. As shown in Figure 5a, the OLEM algorithm has the highest loss rate, which reaches nearly 5% when the number of tasks is 40. However, the other three algorithms, MCORA, OLSM, and ORM, are all below 1%, with MCORA almost 0%. As shown in Figure 5b, when the number of tasks is 40, compared with the OLSM algorithm, OLEM algorithm, and ORM algorithm, the total task completion time of the MCORA algorithm is reduced by 13%, 28%, and 19%, respectively. From Figure 5c, it can be seen that the maximum task time of the OLEM algorithm has a significant upward trend. When the number of tasks is small, the maximum task execution time of the OLEM and ORM algorithm is smaller than that of the OLSM algorithm due to the abundant VECS resources. As the number of TaVs and tasks increases, compared with the OLEM and ORM algorithms, the maximum task completion time of the OLSM algorithm rises slowly. When the number of tasks is greater than 12 and 20, the maximum task completion time of the OLEM algorithm and the ORM algorithm is gradually close to and greater than the OLSM algorithm.

When the task size of TaVs and the computation resource required per bit varied, the cumulative rewards of different algorithms are shown in Figure 6a and Figure 6b, respectively. As the task size and computing density increase, the computation resources required by the TaVs gradually increases, and the cumulative reward decreases.

When the computation capability of vehicles and VECS are changed, the cumulative rewards of different algorithms are shown in Figure 7. It can be seen from Figure 7a that the increases in vehicle computation capability gradually increase the cumulative rewards of all algorithms. The growth trend of these algorithms is similar and stable because they use their computation resources to process tasks, and improving vehicle computing performance is bound to increase the cumulative rewards. However, as shown in Figure 7b, as the VECS performance increases, the cumulative reward of the OLSM algorithm remains unchanged because the OLSM algorithm does not use the VECS resources. The other three algorithms have steadily increased cumulative rewards.

Figure 8a,d, Figure 8b,e, and Figure 8c,f show the comparison of the loss rate, the total task completion time, and the maximum completion time of a single task as the TaVs’ task size and computing density varies, respectively. When the task size or computing density is small, it can be seen from Figure 8a,d that the loss rate of all algorithms is close to 0%. With increased task size or computing density, the loss rate of the OLSM algorithm, OLEM algorithm, and ORM algorithm increases due to insufficient utilization of system computation resources. It can be seen from Figure 8b,e that the growth trend of the total task completion time is similarly under the influence of these two variables.

Meanwhile, as the tasks’ number is 20, the performance of the OLSM and OLEM algorithm is almost the same, which can be confirmed in Figure 4 and Figure 5. When the single task size is 15 Mbits, the total task completion time of the MCORA algorithm is reduced by 18%, 21%, and 20%, respectively, compared with the OLSM algorithm, OLEM algorithm, and ORM algorithm. When the required computation resource per bit is 150 cycles/bit, the total task completion time of the MCORA algorithm is reduced by 20%, 24%, and 20%, respectively, compared with the OLSM algorithm, OLEM algorithm, and ORM algorithm.

Figure 9a–c and Figure 9d–f show the performance comparison with the change in vehicle computing capability and VECS computing capability, respectively. When the computing capacity of the vehicle is small, TaVs and SeVs can only provide fewer computing resources, and the system computing resources are relatively scarce, which leads to the larger loss rate, task completion time, and the maximum single task completion time of the OLSM algorithm, OLEM algorithm, and ORM algorithm. When the vehicle computation capability exceeds 1.5GHz, all algorithms can complete all tasks according to the regulations. When the vehicle computation capability is 3 GHz, the total task completion time of the MCORA algorithm is reduced by 13%, 30%, and 19% compared with the OLSM algorithm, OLEM algorithm, and ORM algorithm, respectively. From Figure 9d, we can see that the loss rate of the OLEM algorithm is less than 1% only when the VECS computation capability is greater than 20 GHz, while the loss rate of the other three algorithms is always in a low range because when the number of tasks is 20, according to the average vehicle computation capability, the computation capability that vehicles can provide is 20 × 15, namely, 30 GHz. It has certain advantages to make full use of ubiquitous vehicle resources reasonably. It can be seen from Figure 9b that when the computation capability of VECS is 30 GHz, the total task completion time of the MCORA algorithm is reduced by 26%, 29%, and 24%, respectively, compared with the OLSM algorithm, OLEM algorithm, and ORM algorithm.

These simulation results show that, compared with the other three schemes, the MCORA scheme can effectively reduce the total delay of task execution, guarantee the QoS of TaVs, and have a certain scalability and stability.

### 5.4. Discussion, Comparison, and Limitations

According to the number of bits each TaV can execute in different modes, the DDPG-based MCORA algorithm is used to select the appropriate task execution mode for each TaV in each time slot τ. Meanwhile, the computing resources of VECS are allocated. Compared with the OLSM and OLEM algorithms, the proposed MCORA algorithm can fully use ubiquitous communication and computing resources in VECNs. Although four execution modes are considered, the simulation results are carried out regarding the number of TaVs, task size, required computation resource per bit, and the computing capability of vehicles and VECS. However, the following shortcomings and limitations still exist: 1. The delay for each task in our proposed system is tmax, regardless of the diversity of tasks; 2. The limited latency of tasks is considered, but the overhead of energy consumption is ignored; 3. DDPG is more challenging to deploy.

## 6. Conclusions

This paper proposes a MCORA optimization model based on DDPG reinforcement learning for the computing task offloading environment of the IoV. Reinforcement learning is used to allocate task offloading modes and VECS computation resources, aiming to solve the problem of the insufficient utilization of system resources in the dynamic environment of VECNs. The proposed method can quickly obtain the approximate optimal solution in a time-varying environment and achieve a low total task completion delay with almost no task lost. The proposed method has better stability and scalability than the existing algorithms. VECNs are gradually developed and improved with the development of cellular networks, and it occupies a certain proportion in the development of 5G and 6G. It can be used for autonomous driving, smart city, and digital twin construction in the future. Additionally, more effective offloading strategies deserve to be formulated by combining task execution and energy consumption because energy saving is essential [38,39,40].

## Figures and Tables

**Figure 1 sensors-24-02205-f001:**
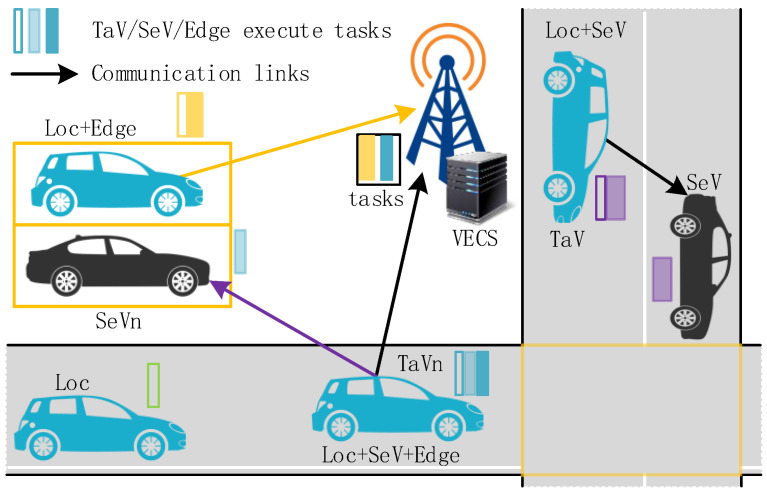
System model.

**Figure 2 sensors-24-02205-f002:**
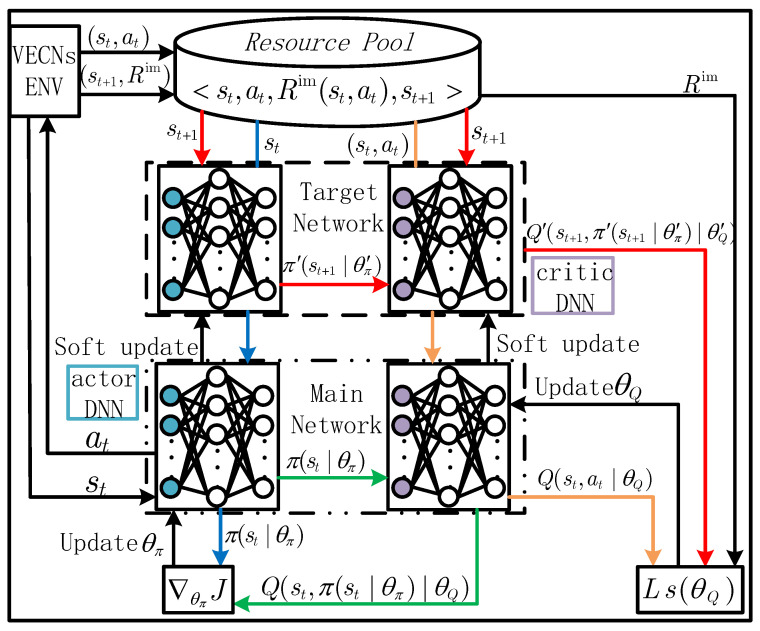
DDPG structure and its update process.

**Figure 3 sensors-24-02205-f003:**
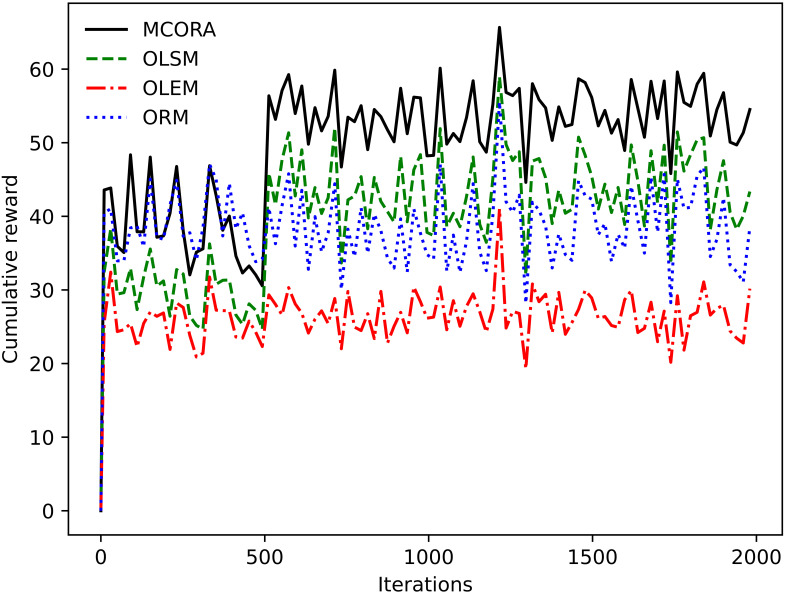
Cumulative rewards of different algorithms vs. the number of iterations.

**Figure 4 sensors-24-02205-f004:**
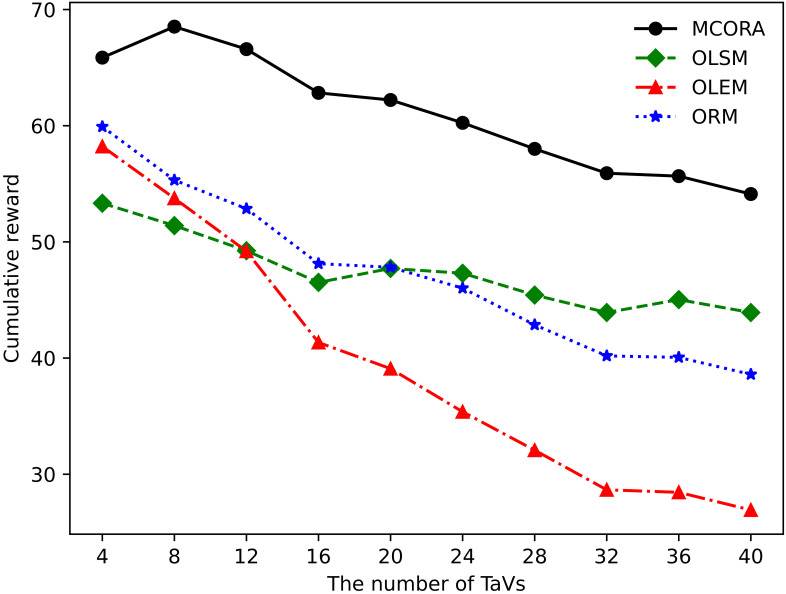
Cumulative rewards of different algorithms vs. the number of TaVs.

**Figure 5 sensors-24-02205-f005:**
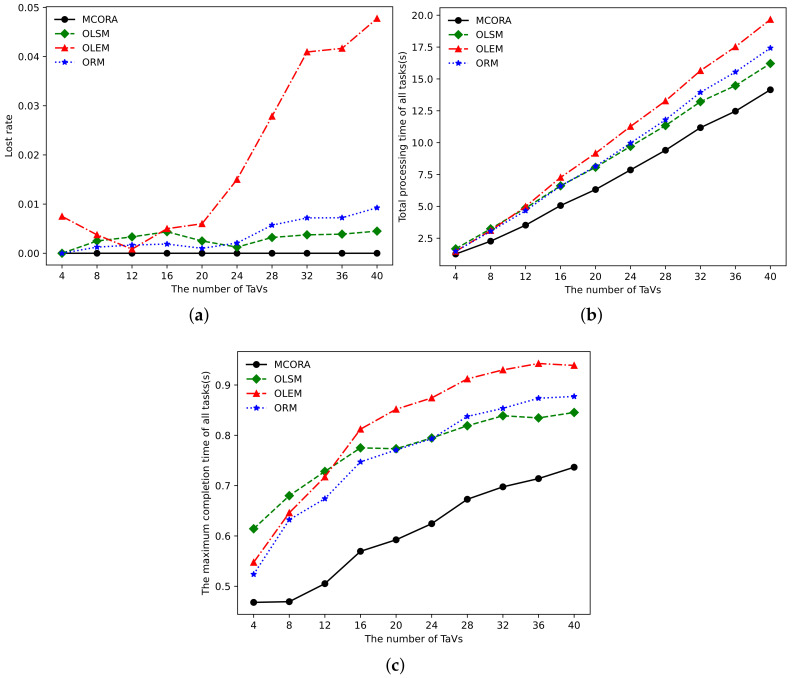
Performance comparison of different algorithms vs. the number of TaVs. (**a**) Loss rate vs. the number of TaVs. (**b**) Total processing time vs. the number of TaVs. (**c**) Tasks’ maximum completion time vs. the number of TaVs.

**Figure 6 sensors-24-02205-f006:**
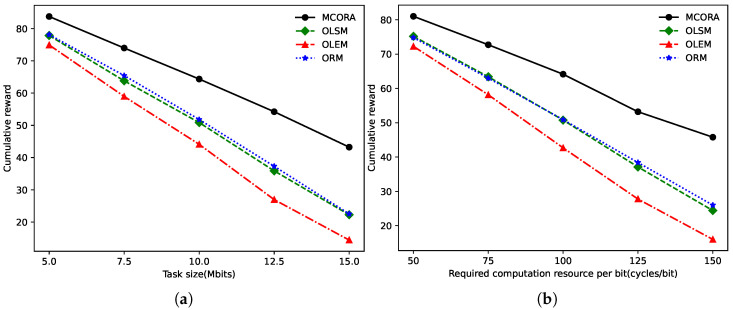
Cumulative rewards of different algorithms as task size and required computation resource per bit vary. (**a**) Cumulative rewards of different algorithms vs. task size. (**b**) Cumulative rewards of different algorithms vs. required computation resource per bit.

**Figure 7 sensors-24-02205-f007:**
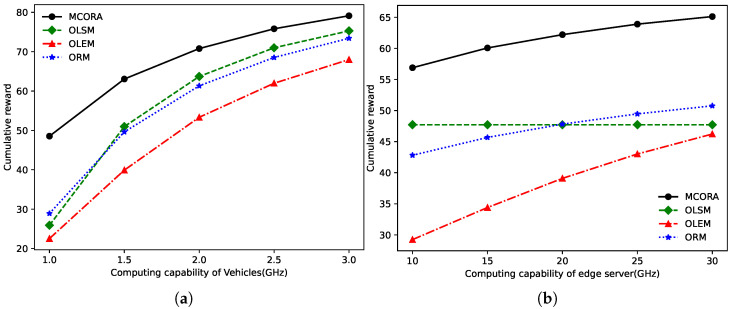
Cumulative rewards of different algorithms as task size and required computation resource per bit vary. (**a**) Cumulative rewards of different algorithms vs. computing capability of TaVs. (**b**) Cumulative rewards of different algorithms vs. computing capability of edge server.

**Figure 8 sensors-24-02205-f008:**
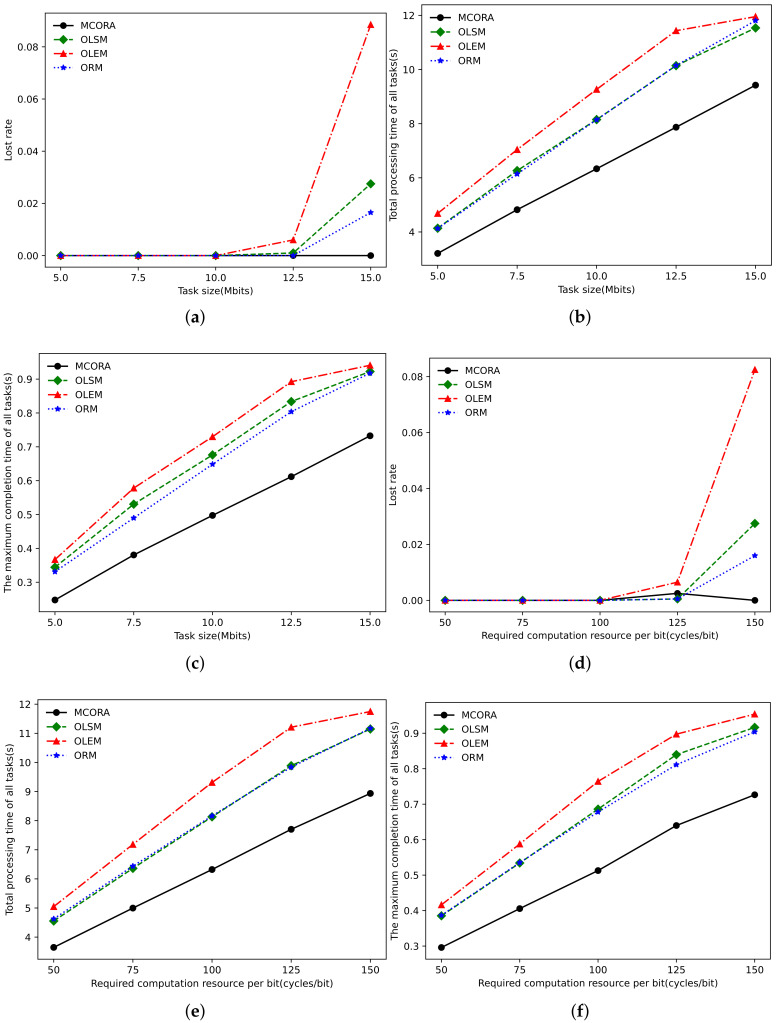
Performance comparison of different algorithms as task size and required computation resources per bit vary. (**a**) Loss rate vs. task size. (**b**) Total processing time vs. task size. (**c**) Tasks’ maximum completion time vs. task size. (**d**) Loss rate vs. required computation resource per bit. (**e**) Total processing time vs. required computation resource per bit. (**f**). Tasks’ maximum completion time vs. required computation resources per bit.

**Figure 9 sensors-24-02205-f009:**
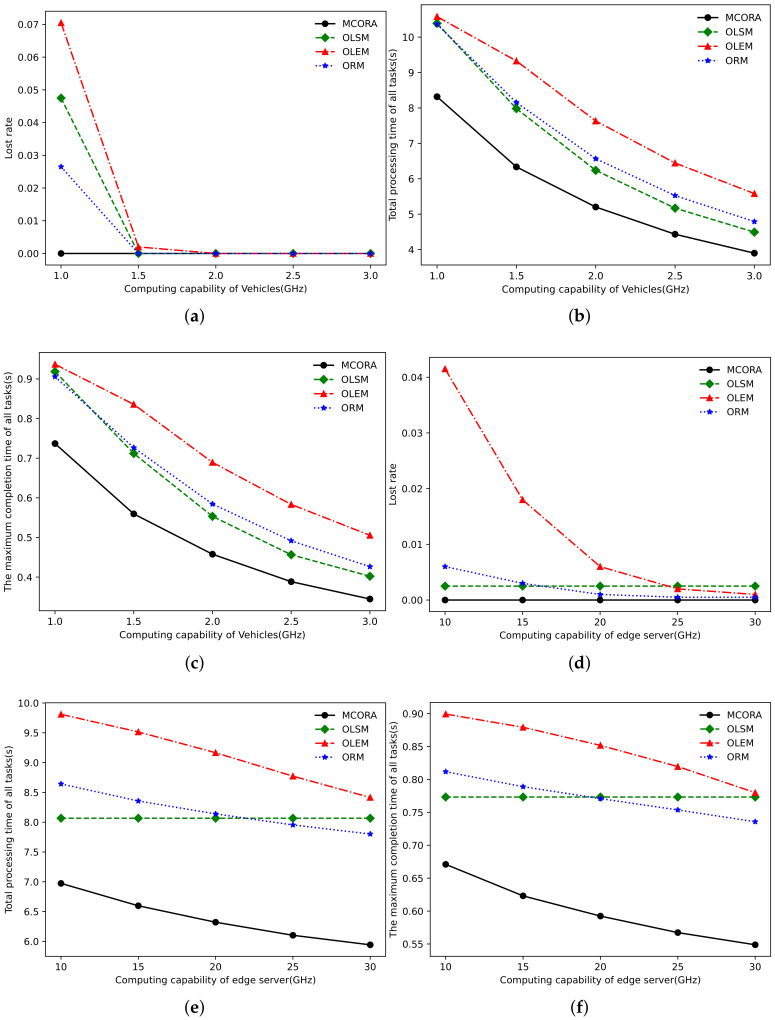
Performance comparison of different algorithms as task size and required computation resource per bit vary. (**a**) Loss rate vs. computing capability of TaVs. (**b**) Total processing time vs. computing capability of TaVs. (**c**) Tasks’ maximum completion time vs. computing capability of TaVs. (**d**) Loss rate vs. computing capability of edge server. (**e**) Total processing time vs. computing capability of edge server. (**f**) Tasks’ maximum completion time vs. computing capability of edge server.

**Table 1 sensors-24-02205-t001:** Comparison with the latest related studies.

Ref.	Year	Mode	Mobility	Method	Advantages	Shortcomings
[9]	2023	$M^{1}$	✓	DDPG	Both computation offloading and power allocation are considered.	Only the resources of the platoon leader are shared.
[10]	2022	$M^{1}$	✓	Q-learning	Jointly consider the cache factor, energy factor, and position factor.	Non-real-time traffic is offloaded into the V2V network.
[14]	2022	$M^{2}$	✗	Deep Q-network	Considering the limited capability of calculating access points and users’ budgets.	The mobility of vehicles is not considered.
[15]	2022	$M^{2}$	✗	Analytical offloading scheme	Multiple computational access points can help vehicular users compute tasks.	Only several scenarios are considered.
[16]	2023	$M^{2}$	✓	DDPG	The trade-off optimization of delay and energy consumption is considered.	Action encoding is used to replace actions in continuous action space.
[19]	2021	$M^{3}$	✓	Heuristics algorithm	Parked vehicles and TaVs driving trajectory prediction are considered.	Each uploaded task is assumed to be performed by only one edge server (0–1 offloading).
[20]	2021	$M^{3}$	✓	Greedy matching	Both the resources of the RSU and nearby vehicles are considered.	0–1 offloading is adopted.
[23]	2023	$M^{3}$	✗	Genetic algorithm	Dispersed computing is considered, including each mobile device, edge, and cloud server.	The solution space dimension is significant, and 0–1 offloading is considered.
[25]	2023	$M^{2}$	✗	Stackelberg game-based scheme	Reasonable prices are designed for computing resources.	Single-server is considered.
[29]	2023	$M^{2}$	✓	Primal-dual DDPG	A multi-tier computation offloading network structure is considered.	The resources of nearby vehicles are not used.
[30]	2023	$M^{2}$	✓	TD3	Considering real-time decision-making and prediction.	0–1 offloading is considered.
[31]	2021	$M^{3}$	✓	DDPG-based	The prioritized experience replay and the stochastic weight averaging mechanisms are considered.	0–1 offloading is considered.
[32]	2022	$M^{3}$	✓	SAC	Both the priority and the size of the tasks are considered.	One TaV and 0–1 offloading is considered.
Proposed		$M^{3}$	✓	DDPG	Several execution modes are considered	The energy consumption is not considered.

**Table 2 sensors-24-02205-t002:** Simulator parameters.

Parameter	Value
Wireless bandwidth of the links (*B*)	2 MHz
The number of TaVs (*N*)	20
Transmit power of TaVn ($P_{n}$)	23 dBm
Noise power ($\delta^{2}$)	−114 dBm
CPU cycle frequency of TaVn ($f_{n}^{\mathrm{loc}}$) or SeVn ($f_{n}^{\mathrm{sev}}$)	[1, 2] GHz
The speed of TaVn ($v_{n}^{\mathrm{tav}}$) or SeVn ($v_{n}^{\mathrm{sev}}$)	[10, 15] m/s
The distance between TaVn and SeVn	[50, 100] m
CPU cycle frequency of the VECS ($F^{edg}$)	40 GHz
Data size of a task ($D_{n}$)	[5, 15] Mbits
The required CPU cycles per bit of a task ($App_{n}$)	[50, 150] CPU cycles/bit

## Data Availability

Data are contained within the article.

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
