# Peer review of "Multi-User Computation Offloading and Resource Allocation Algorithm in a Vehicular Edge Network"

_sensors, 2024, doi:10.3390/s24072205_

Round 1

Reviewer 1 Report

Comments and Suggestions for Authors

Only these remarks are highlighted:

1) Research finding with best result is missing in the abstract.

2) Paper structure at the end of introduction is missing

3) Why only DDPG has been choosen among others RL techniques. Please summarize the existing RL technique in a small flowchat, and justify the chose of using DDPG.

The following ref. maybe benefical: 

Deep Reinforcement Learning for Intrusion Detection in IoT: A Survey

4) There is no mention to which technology this solution is dedicated. Is it for 5G ? This must be mentionned in the manuscript.

5) Perspective is missing, 

Author Response

Thanks for your review. The details are attached.

Reviewer 2 Report

Comments and Suggestions for Authors

multi-user computation offloading and resource allocation optimisation model and algorithm using Deep Deterministic Policy Gradient is an Interesting technique, and the article looks righteous.

Here are few questions”

·         Are the equations used in this paper correct? please provide the references!

·         Lost rate? whether the authors meant the loss rate?.

·         SPAS: Smart Pothole-Avoidance Strategy for Autonomous Vehicles  this
IEEE journal also discusses the DDPG kindly refer this paper as it is also under vehicular networks.

·         System model Fig 1 occupies more white space the unwanted space can be reduced. The figure looks separated meaning the communications among vehicles are not mentioned.

·         The path loss for example equation 1 uses many co-efficients, how those values were attained?

·         Equation 5 looks incomplete

·         Text explanations of figures and equations are mandatory for the clarity of readers

Author Response

(The authors gave the same response as above.)

Reviewer 3 Report

Comments and Suggestions for Authors

1) Considering the complexity of proposed algorithm, a detailed flowchart and architecture diagram illustrating the configuration and steps involved in the proposed study aid readers in understanding the methodology more effectively?

2) The flow of Figure 1 is not clear; the authors must improve the figure and improve the caption of the figure, which gives more insight into the system model.

3) A new section titled "Discussion, Comparison, and Limitations" before the Conclusion enhance the paper's clarity and provide readers with a more comprehensive analysis of the proposed study in relation to existing literature and potential drawbacks?

4) To provide a comprehensive overview of existing research in the field of vehicular computing and optimization, it is pertinent to include recent studies that address various aspects of intelligent vehicular systems, blockchain applications in vehicular networks, and strategies for enhancing energy efficiency in electric vehicles. Therefore, it is recommended to cite the following papers in the introduction and literature review section:

  1. Sohail et al. (2023) present a machine learning-based intelligent vehicular system for driver’s diabetes monitoring in Vehicular Ad-Hoc Networks (VANETs), demonstrating the integration of machine learning techniques in vehicular environments.

  2. Sohail et al. (2023) propose a blockchain-based secure weather forecasting information system through a routing protocol in VANETs, highlighting the potential of blockchain technology in securing data transmission in vehicular networks.

  3. Jamil et al. (2021) introduce PetroBlock, a blockchain-based payment mechanism for fueling smart vehicles, showcasing the utilization of blockchain technology in enhancing vehicular services and transactions.

  4. Naqvi et al. (2024) provide a comprehensive review of development strategies for integrated electronic control units in Internet of Electric Vehicles (IoEVs), offering insights into technological advancements and challenges in improving energy efficiency in electric vehicles.

  5. Nasir et al. (2021) discuss the optimal scheduling of campus microgrids considering the integration of electric vehicles in smart grids, offering valuable perspectives on energy management in the context of electric vehicle integration.

  6. Jamil et al. (2024) analyze driving and braking control logic algorithms for enhancing mobility energy efficiency in electric vehicles, providing insights into improving energy efficiency and performance in vehicular environments.

5) A comparative analysis table in Section 1 of the paper is required, outlining the features, advantages, and limitations of existing studies to help readers better understand the research gap and the uniqueness of our proposed approach.

Author Response

(The authors gave the same response as above.)

Reviewer 4 Report

Comments and Suggestions for Authors

The authors have proposed a multi-user computation offloading and resource allocation algorithm in vehicular edge network. The manuscript is well-organized and needs a minor revision as suggested.

. The introduction should give more explanation about the VECN and multi-user computation offloading and resource allocation processes. [to increase the readability for the naive readers].

. The novelty can be clearly highlighted in the introduction section.

. Another section for related work/ literature survey can be given explaining the existing literature.

. Equation 1 and 2: how the coefficients (numeric values) are considered? can be explained for better readability.

. What is the limitation of this algorithm? Can be explained with the future scope for enhancement.

Some non-technical comments:

. The abbreviations can be put in text also even if mentioned in the abstract.

. There should be a "space" before and after the brackets in the abbreviations.

. Figures are to be put after (not before) their references in the body.

. The equations can be referred to as "(x)" instead of "Equation x".

. Equation 16: spaces between expressions required.

Author Response

(The authors gave the same response as above.)

Round 2

Reviewer 3 Report

Comments and Suggestions for Authors

The author has addressed all the comments. I have no further comments. I am accepting this manuscript.